# DIMENSIONAL DEBIASING VIA MULTI-AGENT CORRECTION

## ABSTRACT

Multimodal Large Language Models (MLLMs) recognize patterns from diverse data dimensions, such as shape, color, and associated language cues. However, inherent biases in training data can lead MLLMs to learn unintended, harmful shortcuts. For example, MLLMs often misinterpret clock times as defaulting to 10:10 due to memorized visual patterns rather than analyzing clock-hand positions. To address this, we propose the Multi-Agent Debiasing (MAD) framework, which performs cross-dimensional verification to correct these shortcut-driven errors. We first derive six dimensions of debiasing guidelines through a systematic analysis of failure responses. These guidelines inform the design of a team of specialized "dimension critic" agents, each an expert in correcting a specific type of error related to either biased cognition or limited perception. In our framework, potentially biased responses are dynamically routed through relevant agents. They then refine and correct the response in cascade over subsequent rounds. We leverage this cascaded correction process as a data engine to build our Multi-Dimensional Debiasing Dataset ($MD^3$), a large-scale collection of rich, debiased reasoning chains. By fine-tuning a model on $MD^3$, we directly teach it to overcome shortcut learning. Our experiments show that the MAD process encourages deeper thinking on biased responses. The MAD framework proves highly effective in classical visual debiasing settings and significantly enhances the reliability of MLLMs.

## 1 INTRODUCTION

Multimodal Large Language Models (MLLMs) Liu et al. (2023); Tong et al. (2024); Radford et al. (2021); Ravi et al. (2024); Chen et al. (2024; 2023); Abdin et al. (2024); Wang et al. (2024) extend Large Language Models (LLMs) Touvron et al. (2023b;a); Abdin et al. (2024); DeepSeek-AI et al. (2025) by incorporating additional input modalities. MLLMs have become increasingly important as they can capture and integrate information across both visual and textual dimensions Zhang et al. (2024a); Wu et al. (2023). However, due to the vastness of their training data, MLLMs often learn from unexpected dataset biases. This occurs when a non-essential attribute becomes spuriously correlated with the target answer, leading the model to adopt a "shortcut"—a simple decision rule based on this spurious correlation. As a result, instead of leveraging their comprehensive reasoning abilities, MLLMs rely on these shortcuts and can be easily misled. For example, Figure 1 demonstrates the "`10:10` dilemma" Deitke et al. (2024). MLLMs often predict the time as `10:10` for any clock image because most clocks in the training data are set to this time, a common practice in advertising for its aesthetically pleasing "V-sign" appearance. The MLLMs learn a harmful shortcut associating clock imagery with a fixed time, ignoring their inherent ability to interpret the positions of the clock hands.

In this paper, we argue that MLLMs adopt such shortcuts because their decision-making lacks comprehensive cross-verification across different dimensions. In the `10:10` dilemma, for instance, a robust model should verify the factual relationship between the visual evidence (the positions of the hour and minute hands) and the textual output (the time). We define this multidimensional verification as a core component of debiased reasoning. A debiasing approach should therefore guide MLLMs away from shortcuts and toward this more principled process.

Inspired by emerging multi-agent systems where complex tasks are decomposed among specialized agents Lu et al. (2022); Feng et al. (2023); Wei et al. (2022); Lightman et al. (2023); Yao et al. (2023); Roucher et al. (2025), we propose the **M**ulti-**A**gent **D**ebiasing (MAD) framework. This

approach uses a team of specialized "dimension critic" agents that work in coordination to perform cross-dimensional verification. Their collaboration systematically reveals and corrects the model's reliance on spurious correlations, fostering a more reliable and multidimensional reasoning process.

The design of these specialized agents is guided by a comprehensive taxonomy of shortcut scenarios we developed to serve as a "shortcut cookbook." This taxonomy categorizes common MLLM failure modes into two primary types: **1**) biased cognition from the LLM component, involving errors in factual and counterfactual reasoning, and **2**) limited perception from the visual encoders, such as confusion over objects, shapes, spatial relationships, distorted OCR, or counting failures. To directly address these perceptual limitations, we enhance our agents by incorporating a visual model zoo with specialized tools like CLIP and Depth Anything.

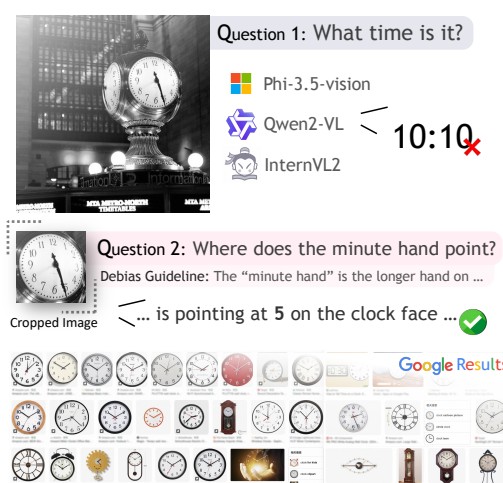

The MAD workflow operates as a cascaded correction system. It begins with a router that triages a biased response to the appropriate agent. The response is then passed through a series of necessary agents, each iteratively correcting a specific error to form a complete, debiased reasoning chain. We leverage this powerful workflow as a data engine to construct the **M**ulti-**D**imensional **D**ebiasing **D**ataset (MD³), which is populated with these rich, corrected reasoning chains. We then use this dataset to fine-tune a model, directly enhancing its resilience to shortcut learning.

Figure 1: **Example of the `10:10` dilemma.** Many state-of-the-art MLLMs default to `10:10` by memorizing a common pattern rather than reasoning about the positions of the clock hands, a shortcut learned from biased internet images.

We validated the effectiveness of our fine-tuned model through extensive experiments. Our model shows consistent improvements on general MLLM benchmarks like AI2 Diagrams and on shortcut-specific evaluations such as OCRBench, RealWorldQA, and CV-Bench. Furthermore, it achieves substantial progress on datasets with pronounced biases, including VQA-CP, VQA-CE, and GQA-OOD, proving that our cascaded multidimensional debiasing workflow is effective even with less robust visual models. Our contributions are thus threefold: the proposal of the novel MAD framework, the development of a shortcut taxonomy and the resulting MD³ dataset, and the demonstration of our approach's effectiveness in significantly improving model robustness and reasoning.

Our contributions are:

- A novel Multi-Agent Debiasing (MAD) framework that employs specialized critic agents to correct shortcut biases across multiple reasoning dimensions.

- A comprehensive taxonomy of MLLM shortcut types that guides the creation of our new Multi-Dimensional Debiasing Dataset (MD³), a large-scale resource of debiased reasoning chains.

- Extensive experiments showing that models fine-tuned on MD³ achieve substantial gains in robustness and reasoning capabilities across a wide range of MLLM evaluations.

## 2 PRELIMINARY

In this section, we start by exploring how machine learning models capture data dimensions. We define three types of dimensions within the dataset bias view. By revisiting real-world datasets, we define shortcuts and discuss related work that explains why models tend to learn shortcuts and what debiasing means in MLLM.

**Dimensions and their relationship with target objects.** Real-world data can often be described through various dimensions (attributes) $d$. With Figure 2, we use *shape*, *color*, and *text on boat* as examples of the dimensions Zhang et al. (2023).

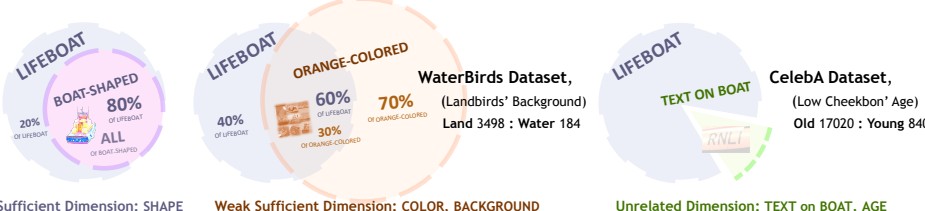

Figure 2: **Examples of Venn diagrams** illustrating the count distribution of sufficient, weak sufficient, and unrelated dimensions.

- Some dimensions are strongly associated with the object, allowing identification based solely on these features, *i.e.*, the shape and function of a lifeboat are distinctive enough to identify it.
- Other dimensions have a weaker association. While most target objects may have these features, they are not exclusively sufficient to identify the object, *i.e.*, although most lifeboats are *orange*, not all *orange* objects are lifeboats. However, the *orange* color can suggest that a boat might be a lifeboat, as this color is often used for lifeboats.
- Finally, some dimensions are unrelated, containing features that have no bearing on identification, *i.e.*, image watermarks or boat numbering do not impact the identification of a lifeboat.

Based on the above properties, we define the dimensions conditional on the target object.

**Definition 1.** Dimensions in Dataset Bias View.

- Sufficient dimensions (*satisfy all*),

$$\mathbf{x} \in \mathcal{F}\left(d_{\text{sufficient}}\right) \Rightarrow \mathbf{x} \text{ is target object .}$$

- Weak sufficient & unrelated dimension,

$$\mathbf{x} \in \mathcal{F}\left(d_{\text{weak-sufficient}}\right) \not\Leftrightarrow \mathbf{x} \text{ is target object .}$$

  – The weak one, and unrelated one,

$$\left|\mathcal{F}\left(d_{\text{weak-sufficient}}\right) \cap \boldsymbol{X}_{\text{of target}}\right| \approx \left|\boldsymbol{X}_{\text{of target}}\right| ; \ \ \mathcal{F}\left(d_{\text{unrelated}}\right) \cap \boldsymbol{X}_{\text{of target}} \sim \text{Uniform Dist. ,}$$

where $\mathcal{F}\left(\cdot\right)$ represents a particular feature in dimension, such as *boat-shaped*, *orange*, or a specific *watermark*, meanwhile, the $\boldsymbol{X}_{\text{of target}}$ refers to sub-dataset of target objects.

**Revisit the real-world distribution of dimensions.** Following the Definition 1, Figure 2 uses a Venn diagram Ho et al. (2021) to illustrate the distribution of dimensional characteristics. Most samples of lifeboat have both sufficient dimensions (*e.g.*, being *boat-shaped*, *inflatable*, and *having life rings*) and weak sufficient dimensions (*e.g.*, being *orange*). However, some examples, such as "life rafts" or "platforms", may differ in color. Thus, the *orange* cannot be a sufficient characteristic, as not all *orange* items are lifeboats. Irrelevant dimensions should be randomly distributed since they have no relation to the object. Some bias-synthetic datasets Li et al. (2023); Sagawa et al. (2020a) leverage unrelated dimensions to introduce training biases. For instance, in the CelebA Liu et al. (2015) dataset, Age is an unrelated dimension when predicting whether cheekbones are *high* or *low*. The training bias arises if the dataset contains numerous samples where the same age group is consistently paired with a particular cheekbone characteristic. Additionally, some datasets Agrawal et al. (2017); Dancette et al. (2021) introduce bias on weak sufficient dimensions as real-world scenarios. For example, in the Waterbirds Sagawa et al. (2020a) dataset, most objects on a water background are waterbirds instead of landbirds.

**Shortcut with spurious correlation.** When a training set includes many samples with a dimension spuriously correlated to the target, a shortcut can develop, allowing the model to make decisions based solely on these weak-sufficient or unrelated dimensions. For instance, if most lifeboats in a training set are *orange*, the model may rely on weak sufficient dimension color to recognize lifeboats. While this may achieve high accuracy within the training set, it proves useless and harmful for general decision-making.

**Related works.** Some works Hermann et al. (2024); Nicolicioiu et al. (2023); Pezeshki et al. (2021); Hermann & Lampinen (2020) indicate that models often rely on shortcuts because features in weak-sufficient dimensions are typically easier-to-learn than those in essential dimensions as they prioritize

**Why** bias?

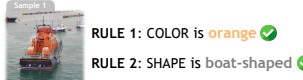 RULE 1: COLOR is orange ✅
RULE 2: SHAPE is boat-shaped ✅

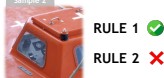 RULE 1 ✅
RULE 2 ❌

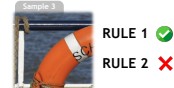 RULE 1 ✅
RULE 2 ❌

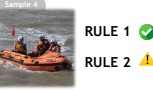 RULE 1 ✅
RULE 2 ⚠️

If the model only discriminates the *lifeboat* based on the *color* dimension (RULE 1), then bias is coming

Figure 3: **The forming of shortcut**. How MLLMs learn color-based shortcuts: *orange* dominance in the training set, while neglecting harder dimensions, like *shape* or other ones.

fitting the former. Further studies examine how different model architectures Izmailov et al. (2022), parameter intensities McAleese et al. (2024), and the alignment Cheng et al. (2021) or decoupling Kim et al. (2021) of features impact the extent of shortcut learning. In our paper, we address the bias problem in sophisticated MLLMs with extensive parameters. Traditional debiasing techniques may become impractical due to significant storage and computational demands at the feature-capturing level. Zhang et al. proposed a "post-hoc" debiasing decoding method using nonsensical input; however, the unbiasedness derived from nonsensical knowledge might inadvertently discard helpful information from what we term weak sufficient dimensions. Other methods Lim et al. (2023) employ strong supervision signals by creating adversarial samples from out-of-domain data to regulate model responses. Our MAD workflow highlights the reliance on shortcuts by lacking consideration of additional dimensions and uses guided debiasing via a data engine to improve the reasoning process for reliable debiasing. Addressing bias in MLLMs is crucial for their application in specialized domains, managing hallucinations Bai et al. (2024) and refusals Mazeika et al. (2024); Shao et al. (2024b), and ensuring safety in broader societal contexts Gallegos et al. (2024).

## 3 THE MAD FRAMEWORK

In this section, we first provide an overview of our Multi-Agent Debiasing (MAD) framework. We then introduce our "shortcut cookbook", a taxonomy of common MLLM failure modes, and explain how it guides the design of our specialized agents and the construction of our debiasing dataset.

### 3.1 A MULTI-AGENT APPROACH TO DEBIASING

**Motivation:** The core problem with shortcut learning is that a model's decision-making process is one-dimensional, relying on a single spurious correlation while ignoring other critical evidence. To counter this, a debiasing method must enforce a more holistic, multidimensional verification process. It should compel the model to cross-reference different facets of the input, ensuring that its conclusion is supported by consistent evidence from multiple dimensions, such as textual semantics, object relationships, and spatial logic.

Inspired by the success of multi-agent systems in decomposing complex problems, our MAD framework operationalizes this principle of multidimensional verification. We replace the monolithic reasoning process of a single MLLM with a collaborative workflow of specialized agents. Each agent, or "dimension critic," is an expert in identifying and correcting a specific type of shortcut bias. By working in concert, these agents systematically analyze a problem from multiple angles, replacing a flawed, shortcut-based conclusion with a robust, well-reasoned one. This agent-based collaboration effectively simulates the comprehensive reasoning process that debiased models should ideally perform internally.

**Overall Workflow.** As illustrated in Figure 3, the MAD workflow begins with an initial, potentially biased response from a base MLLM. This response is first sent to a **Router Agent**, denoted as $\mathcal{R}$, which acts as a dispatcher. The router's task is to analyze the response and identify the primary dimension of the shortcut error. Based on its diagnosis, it routes the problem to the appropriate specialized **Dimension Critic Agent**, $\mathcal{A}_k$, where $k$ indexes the agent's area of expertise, *e.g.*, factual reasoning, spatial analysis, etc.

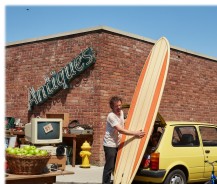

**How** do Multi-Agent on Multi-Dimension to Debias?

**Factual Reasoning:** The surfboard cannot fit into the car.

**Counterfactual Verification:** a fire hydrant doesn't have to be red, and an orange surfboard isn't a lifeboat.

**Small, Edge, OoD Objects:** The partially seen tricycle is an Edge object; the computer at a fruit stand is OoD.

**Spatial Relations & Counting:** Challenges include understanding the person-surfboard-car interaction and counting the stacked, occluded apples.

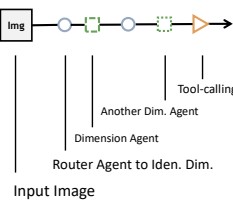

Figure 4: **The cascaded workflow** of the Multi-Agent Debiasing (MAD) framework.

The selected agent then corrects the specific error, producing a refined reasoning step. If the router detects multiple types of errors, it can initiate a cascaded correction sequence, where the output from one agent becomes the input for the next. This creates a complete, debiased reasoning chain.

We can formally represent this cascaded agent workflow. Let $\mathbf{x}$ be the multimodal input and $F(\mathbf{x})$ be the initial, biased response from a base model $F$. Let the set of $K$ specialized critic agents be $\{\mathcal{A}_1, \mathcal{A}_2, \ldots, \mathcal{A}_K\}$. The process unfolds as follows:

$$
\begin{aligned}
\mathbf{y}_0 &= F(\mathbf{x}) \quad \text{// Initial biased response} \\
k_1, k_2, \ldots, k_n &= \mathcal{R}(\mathbf{y}_0) \quad \text{// Router identifies sequence of agents} \\
\mathbf{y}_i &= \mathcal{A}_{k_i}(\mathbf{y}_{i-1}) \quad \text{// Agent } i \text{ corrects the output from the previous step} \\
\mathbf{y}_{\text{final}} &= \mathcal{A}_{k_n} \circ \cdots \circ \mathcal{A}_{k_2} \circ \mathcal{A}_{k_1}(\mathbf{y}_0)
\end{aligned}
\tag{1}
$$

Here, the symbol $\circ$ denotes function composition, illustrating how agents build upon each other's corrections to form the final debiased response, $\mathbf{y}_{\text{final}}$. The objective of MAD is not just to get the right answer but to produce a high-quality, transparent chain of debiasing thoughts. This rich output is then used to construct our training data.

### 3.2 A Shortcut Cookbook for Agent Specialization

**Motivation:** To design effective specialized agents, we first needed a systematic understanding of the errors they must correct. Manually annotating these diverse failures is prohibitively expensive and time-consuming. Therefore, we developed a **Shortcut Cookbook**, a comprehensive taxonomy that categorizes common MLLM failure modes. This cookbook serves as a blueprint for designing our agents' roles and for creating guidelines—analogous to an "AI Constitution" Bai et al. (2022)—to regulate our automated data annotation process. We group the shortcuts into two primary categories, which guide the specialization of our critic agents.

**1. Biased Cognition (Errors from the LLM component):** These agents address failures in reasoning that are not purely perceptual.

- **Factual Reasoning:** The model correctly identifies objects but fails to reason about their factual relationships or properties, *e.g.*, knowing a fire hydrant is for water but failing to connect it to a firefighter in the image.
- **Counterfactual Verification:** The model defaults to a memorized stereotype or common correlation, even when visual evidence contradicts it, *e.g.*, stating a banana is yellow when the image shows a green one.

**2. Limited Perception (Errors from the Vision component):** These agents tackle failures in visual understanding, often enhanced with specialized vision tools.

- **Object and Attribute Recognition:** The model overlooks small, peripheral, or out-of-domain objects, focusing only on the most salient elements.
- **OCR and Text Understanding:** The model struggles to read text that is distorted, occluded, rotated, or contextually complex.
- **Spatial Relationships:** The model identifies multiple objects but misinterprets their relative positions, sizes, or interactions, *e.g.*, saying an object is "on" a table when it is "under" it.
- **Counting:** The model fails to accurately count the number of objects, *e.g.*, in cluttered scenes.

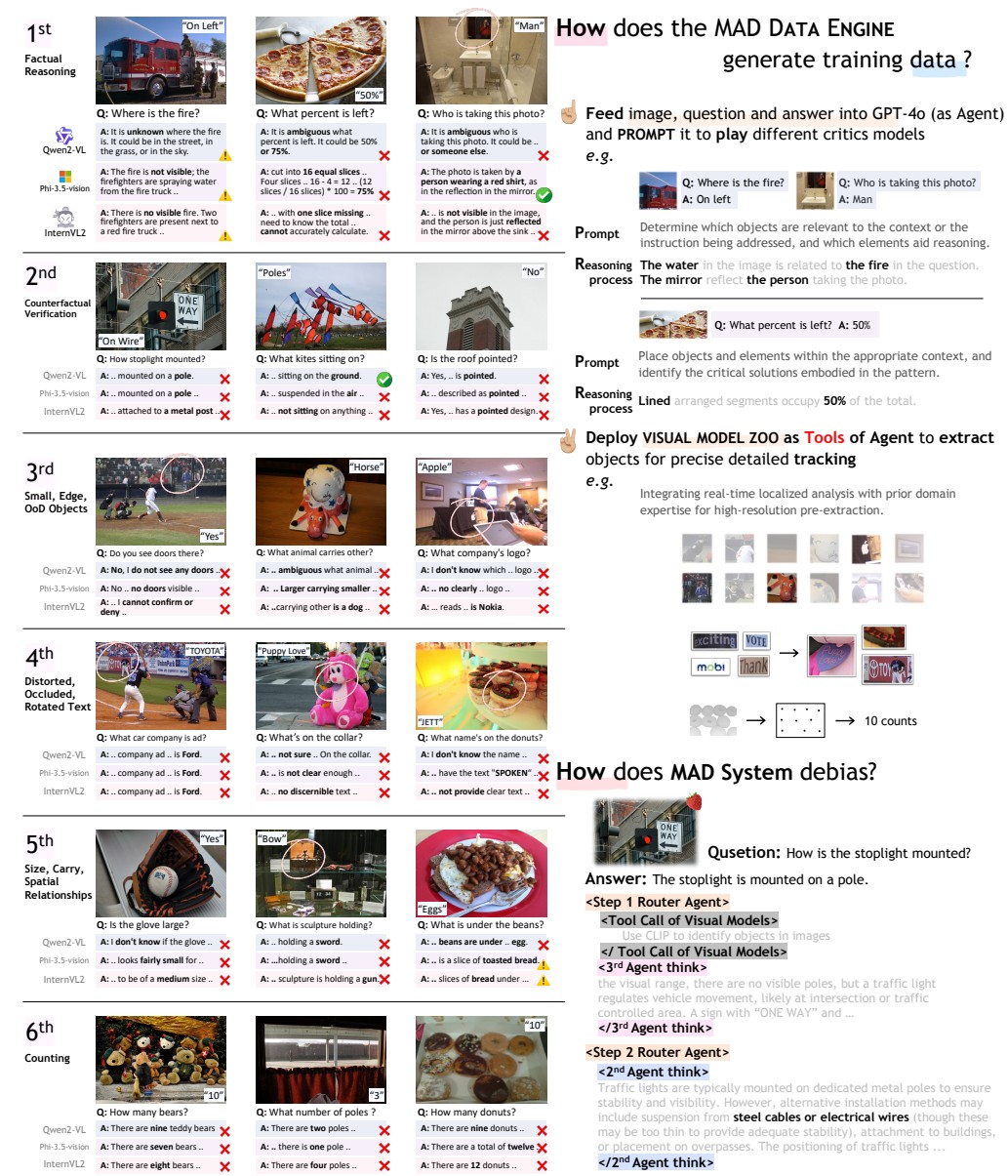

Figure 5: **Illustration of Shortcut Cookbook. 1**) The left part demonstrates samples of different bias types. **2**) Right part: we illustrate how to prompt to generate a debiased reasoning process and how MAD with critic agents to debias.

As illustrated in Figure 5, these failure patterns are consistent across various state-of-the-art MLLMs and datasets like VQA-CP Agrawal et al. (2017), VQA-CE Dancette et al. (2021), and GQA-OOD Kervadec et al. (2020). Our cookbook provides a structured way to address these fundamental gaps in multidimensional reasoning.

### 3.3  THE MD³ DATASET

**Motivation:** Using the MAD framework, we built an automated data engine to generate the **Multi-Dimensional Debiasing Dataset (MD³)**. Simply using a powerful MLLM for annotation risks propagating the very biases we aim to eliminate. Therefore, our engine integrates the MAD agent-based workflow with a suite of specialized tools to ensure the generated reasoning chains are reliable.

| Model / Dataset | LLaVA-v1.5-7B | | | LLaVA-Llama-3-8B | | | Llama-3.2-11B-Vision | | |
|---|---|---|---|---|---|---|---|---|---|
| | Base | MAD trained | Impro. | Base | MAD trained | Impro. | Base | MAD trained | Impro. |
| *General Knowledge, Comprehension, and Reasoning* | | | | | | | | | |
| MMMU Yue et al. (2024) (Val) | 35.07 | 35.41 | 0.34 | 36.76 | 36.09 | -0.67 | 48.76 | 47.17 | -1.59 |
| AI2 Diagrams Kembhavi et al. (2016) (Test) | 54.95 | 56.12 | 1.17 | 60.52 | 60.98 | 0.46 | 77.23 | 76.17 | -1.06 |
| ScienceQA Lu et al. (2022) (Test) | 71.89 | 71.74 | -0.15 | 74.27 | 74.71 | 0.44 | 77.05 | 77.39 | 0.34 |
| *Subgroups with Potential Bias* | | | | | | | | | |
| MMBench Liu et al. (2024a) | 66.41 | 67.75 | 1.34 | 69.23 | 71.77 | 2.54 | 72.46 | 73.92 | 1.46 |
| OCRBench Liu et al. (2024b) | 320 | 436 | 116 | 415 | 472 | 57 | 753 | 745 | -8 |
| RealWorldQA X.ai (2024) | 53.86 | 57.91 | 4.05 | 56.21 | 59.61 | 3.40 | 59.08 | 61.44 | 2.36 |
| CV-Bench Tong et al. (2024) (2D) | 55.56 | 61.89 | 6.33 | 61.68 | 63.49 | 1.81 | 64.05 | 64.19 | 0.14 |
| (3D) | 57.00 | 58.17 | 1.17 | 63.42 | 66.17 | 2.75 | 66.33 | 66.67 | 0.34 |
| Hallusion Guan et al. (2024) (VD) | 53.13 | 56.51 | 3.38 | 48.90 | 54.99 | 6.09 | 53.81 | 55.33 | 1.52 |
| (VS) | 43.89 | 57.78 | 13.89 | 61.94 | 56.39 | -5.55 | 56.11 | 62.22 | 6.11 |
| (All) | 49.63 | 56.99 | 7.36 | 53.84 | 55.52 | 1.68 | 54.68 | 57.94 | 3.26 |
| *Challenging Biased Scenarios* | | | | | | | | | |
| VQA-CP Agrawal et al. (2017) (Test) | 63.98 | 68.83 | 4.85 | 68.59 | 69.69 | 1.10 | 70.86 | 74.14 | 3.28 |
| VQA-CE Dancette et al. (2021) (Hard) | 58.73 | 63.35 | 4.62 | 60.09 | 62.04 | 1.95 | 64.30 | 64.27 | -0.03 |
| GQA-OOD Kervadec et al. (2020) (Tail) | 59.36 | 63.78 | 4.42 | 63.95 | 65.10 | 1.15 | 68.11 | 69.99 | 1.88 |
| MD³ (Ours) | 40.72 | 63.21 | **22.49** | 46.11 | 66.88 | **20.77** | 52.83 | 67.80 | **14.97** |

Table 1: **Performance comparisons of MLLM trained by MAD data engine** on the general, emphasizing spurious and highly biased multimodal instruction sets. We apply the generated dataset to three base MLLMs: LLaVA-v1.5-7B, LLaVA-Llama-3-8B, and Llama-3.2-11B-Vision. In each architecture, the third column is the performance improvement, with darker indicating greater one.

- **Cognitive Agents (for Factual & Counterfactual Errors):** To correct reasoning biases, we use a powerful annotation MLLM (GPT-4o$_{mini}$ OpenAI et al. (2024)) prompted with fine-grained object descriptions. By providing detailed context and using reason-based prompts, we guide the agent to analyze inter-object relationships and underlying factual semantics, rather than relying on shortcuts.
- **Perception Agents (for Visual Errors):** To address perceptual limitations, we enhance our agents with a **Visual Model Zoo**. This includes specialized tools like CLIP Radford et al. (2021) for classification, SAM 2 Ravi et al. (2024) for segmentation, and Depth Anything V2 Yang et al. (2024) for 3D estimation. Following a Visual-CoT Wu & Xie (2024); Shao et al. (2024a) approach, we feed the outputs of the vision tools back to the annotation MLLM with dimension-specific prompts, enabling it to correct its initial perceptual errors.

The complete workflow of our MD³ data engine operates as follows:

1. **Cold-Start Phase:** We initialize the Router Agent using existing dataset annotations to provide an initial classification of error types for failed MLLM responses.

2. **Agent-Based Correction:** For each failed response, the Router Agent dispatches it to the appropriate critic agent based on the Shortcut Cookbook.

   i. Cognitive failures trigger our reason-based prompting protocol.

   ii. Perceptual failures activate the Visual Model Zoo pipeline.

3. **Validation & Iteration:** We verify if the agent's generated reasoning chain successfully mitigates the bias. Approved chains are added to the MD³ dataset. Both successfully corrected and persistently biased examples are used as positive and negative samples to further train the Router Agent, improving its diagnostic capabilities over time.

Ultimately, guided by our Shortcut Cookbook, this data engine leverages the MAD workflow to systematically generate high-quality training data for both the **router** and the **dimension critic agents**. The resulting dataset, rich with debiased reasoning processes, is then used to fine-tune an MLLM.

# 4 EXPERIMENTS

This section outlines the evaluation, data construction, and training details of MAD, followed by an analysis of its performance improvements across various scenarios.

## 4.1 TOWARDS RELIABLE DEBIASING PERFORMANCE

**Details of the benchmark in Table 1.** We evaluate the general knowledge, comprehension, and reasoning datasets, such as MMMU Yue et al. (2024), AI2Diagrams Kembhavi et al. (2016), and ScienceQA Lu et al. (2022). These datasets focus on various general domains featuring multiple types of visual instructions and answers. Additionally, we assess datasets that emphasize spurious correlation, including MMBench Liu et al. (2024a), OCRBench Liu et al. (2024b), RealWorldQA, and CV-Bench Tong et al. (2024). MMBench covers perception and reasoning aspects, OCRBench deals with text recognition, RealWorldQA involves real-world spatial relationships, and CV-Bench addresses visual object relationships, counting in 2D, and distance or proximity in 3D. Lastly, we examine classic debiased visual question-answering datasets: VQA-CP v1 test Agrawal et al. (2017), VQA-CE hard Dancette et al. (2021), and GQA-OOD tail Kervadec et al. (2020). From VQA-CP, we sample approximately 200 examples for each type of shortcut dimension, totaling 1,280 samples in the dataset.

**Details of base biased MLLM.** In Table 1, we examine the performance of MAD across three distinct base model architectures: LLaVA-v1.5-7B Liu et al. (2023), LLaVA-Meta-Llama-3-8B Liu et al. (2023); AI@Meta (2024), and Llama-3.2-11B-Vision Touvron et al. (2023b;a). These models showcase different intrinsic biases inherent in their architectures. LLaVA-v1.5-7B and LLaVA-Meta-Llama-3-8B are trained

| Model / Dataset | Base | MAD Agent Num. | | |
|---|---|---|---|---|
| | | 2 / 6 | 4 / 6 | 6 / 6 |
| Hallusion (All) | 49.63 | 50.79 | 54.68 | 56.99 |
| VQA-CP (Test) | 63.98 | 65.17 | 66.49 | 68.83 |
| GQA-OOD (Tail) | 59.63 | 61.57 | 62.45 | 63.78 |
| $MD^3$ (Ours) | 40.72 | 53.24 | 60.02 | 63.21 |

Table 2: **Ablation studies.** Biased outputs are increasingly corrected as the number of activated agents increases.

using the pretraining and instruction tuning data provided by LLaVA-v1.5 Liu et al. (2023). In contrast, the Llama-3.2-11B-Vision is aligned with its publicly released version.

**Details of the $MD^3$ Data Engine.** Our MAD data engine is orchestrated by GPT-4o, which serves as both the intelligent router agent and the core of the dimension critic agents during the annotation phase. The process begins by collecting erroneous and shortcut-driven responses from the base biased MLLMs. Each biased output is then processed through the MAD workflow: the router agent first diagnoses the failure type and dispatches it to the appropriate specialized critic agent. The critic, guided by our shortcut cookbook and enhanced with the visual model zoo, generates a corrected, multi-step reasoning chain. This entire process transforms simple biased outputs into rich, labeled training samples for our $MD^3$ dataset.

**Details of Fine-tuning.** Our final $MD^3$ dataset consists of approximately $50k$ debiased reasoning chains, generated from an initial pool of $90k$ instruction prompts. During instruction tuning, this debiasing data is mixed with a general instruction fine-tuning set in a $1:3$ ratio. Following the LLaVA-v1.5 Liu et al. (2023) methodology, we conduct a two-stage fine-tuning process. In the first stage, only the visual connector is trained with a learning rate of $1e^{-3}$. In the second stage, we fine-tune both the LLM and the connector for one epoch, with learning rates of $2e^{-6}$ and $1e^{-5}$, respectively. All models are trained on $8 \times$ A100 GPUs.

**Details of MAD for classic visual debiasing tasks.** We also validate our approach in classic visual debiasing settings by using an ImageNet Deng et al. (2009) pre-trained ResNet50 He et al. (2016) as our vision backbone. First, we train a base model with empirical risk minimization on the biased training set. Then, we implement agents that act as linear correctors, focusing on either class or semantic aspects based on features from the backbone. During inference, the corrective outputs from the agents are combined with the base model's prediction to produce a refined, debiased result. Image features are extracted once, allowing the agent corrections to be stacked efficiently. Additional training hyperparameters are detailed in the Appendix.

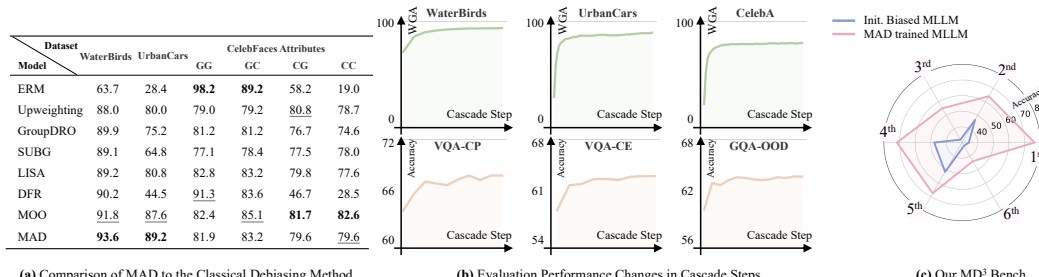

| Dataset | WaterBirds | UrbanCars | CelebFaces Attributes | | | |
|---|---|---|---|---|---|---|
| Model | | | GG | GC | CG | CC |
| ERM | 63.7 | 28.4 | **98.2** | **89.2** | 58.2 | 19.0 |
| Upweighting | 88.0 | 80.0 | 79.0 | 79.2 | 80.8 | 78.7 |
| GroupDRO | 89.9 | 75.2 | 81.2 | 81.2 | 76.7 | 74.6 |
| SUBG | 89.1 | 64.8 | 77.1 | 78.4 | 77.5 | 78.0 |
| LISA | 89.2 | 80.8 | 82.8 | 83.2 | 79.8 | 77.6 |
| DFR | 90.2 | 44.5 | 91.3 | 83.6 | 46.7 | 28.5 |
| MOO | 91.8 | 87.6 | 82.4 | 85.1 | **81.7** | **82.6** |
| MAD | **93.6** | **89.2** | 81.9 | 83.2 | 79.6 | 79.6 |

**(a)** Comparison of MAD to the Classical Debiasing Method    **(b)** Evaluation Performance Changes in Cascade Steps    **(c)** Our MD³ Bench

Figure 6: Performance comparison of classic vision tasks, cascade step variation, and MD³ bench, . (**a**) We compare different debiasing methods, including Upweighting, GroupDRO Sagawa et al. (2020a), SUBG Sagawa et al. (2020b), LISA Yao et al. (2022), DFR Kirichenko et al. (2023a), and MOO Kim et al. (2024), based on the pretrained ResNet-50. Different groups is divided by sufficient or weak-sufficient dimensions. Worst Group Accuracy (WGA) is measured on WaterBirds and UrbanCars, while CelebFaces Attributes analysis covered each group, with 'G' correlating the weak-sufficient dimension and 'C' being the sufficient one. **b**) Performance growth with cascade steps. We plot curves for each cascade step in the experiments from Table 1 and part (a). **c**) We evaluate the multi-dimensional debiasing dataset with LLaVA v1.5 (in light blue) and our correction of it. From 1ˢᵗ to 6ᵗʰ correspond to the six dimensions in Figure 5.

**Main Results.** In Table 1, we analyze how MAD reduces biases in three MLLM architectures. Although these base models often perform well, they still exhibit significant biases when faced with spurious correlations. Our study shows that MAD effectively mitigates these biases, improving accuracy by over 4% on challenging benchmarks like VQA-CP Agrawal et al. (2017), VQA-CE Dancette et al. (2021), GQA-OOD Kervadec et al. (2020), and RealWorldQA X.ai (2024). In addition to reducing bias, MAD also reduces hallucination Bai et al. (2024) issues.

**Ablation studies.** In Table 2, we evaluate the utility of different critic agents. The notation "2 / 6" indicates using agents for *Factual Reasoning* (1ˢᵗ) and *Small, Edge, OoD Objects* (3ʳᵈ), while "4 / 6" adds agents for *Counterfactual Verification* (2ⁿᵈ) and *Size, Carry, Spatial Relationships* (5ᵗʰ). As more specialized agents are incorporated, responses undergo more comprehensive iterative refinement. Even interventions from just two agents yield significant performance boosts on GQA-OOD and our MD³ benchmark.

**Debiasing on classic vision tasks.** In Figure 6, we compare MAD with several debiasing methods (*e.g.*, GroupDRO Sagawa et al. (2020a), SUBG Sagawa et al. (2020b), LISA Yao et al. (2022), and DFR Kirichenko et al. (2023a)) on ResNet-50, reporting the Worst Group Accuracy (WGA) on WaterBirds and UrbanCars, and per-group accuracy on CelebFaces Attributes. We then illustrate that performance consistently grows with an increasing number of agent correction steps, plotting the corresponding improvement curve.

## 5 Discussion and Conclusion

In this paper, we introduced MAD (Multi-Agent Debiasing), a novel framework that effectively mitigates shortcut biases in MLLMs by employing a collaborative team of specialized agents. Instead of relying on a single, monolithic reasoning process, MAD decomposes the debiasing task across "dimension critic" agents, each an expert in a specific type of MLLM failure. Through a cascaded correction workflow, our approach guides MLLMs away from spurious correlations and toward a more comprehensive, multidimensional reasoning process. Our comprehensive shortcut taxonomy and the automated data engine, which produced the Multi-Dimensional Debiasing Dataset (MD³), provide a scalable methodology for improving MLLM robustness. The MAD framework is inspired by the broader trend of multi-agent systems and verifiable, step-by-step reasoning Cobbe et al. (2021); Lightman et al. (2023); McAleese et al. (2024); DeepSeek-AI et al. (2025). By applying this collaborative approach specifically to the problem of debiasing, we highlight its potential to significantly enhance the reliability and trustworthiness of MLLMs in various applications.

ETHICS STATEMENT

We have strictly adhered to the ICLR Code of Ethics. Our research does not involve any human subjects, sensitive data, or personally identifiable information. The work presented in this paper does not raise concerns regarding discrimination, bias, or potential for malicious use. We have conducted our research with integrity and are committed to the responsible advancement of machine learning.

REPRODUCIBILITY STATEMENT

We are committed to ensuring the reproducibility of our research. All implementation details, model architectures, and hyperparameters are thoroughly described in the main body of the paper and the appendix. To further facilitate reproducibility, we will make our complete source code and experimental scripts publicly available in a GitHub repo. upon the acceptance.

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

# A APPENDIX

## A.1 INTRODUCTION

This research aims to address a critical issue in Multimodal Large Language Models (MLLMs): their tendency to rely on "shortcut features" instead of learning robust and generalizable representations. These shortcuts are spurious correlations in the training data that allow the model to achieve high accuracy on the training distribution but fail in real-world scenarios. This phenomenon causes models to over-focus on simplistic cues while neglecting a comprehensive, multi-dimensional analysis of the input. To systematically tackle this challenge, our main paper introduces the Multi-Agent Debiasing (MAD) framework. This appendix elaborates on our methodology, including the completeness of our bias taxonomy (the "Shortcut Cookbook"), the construction details of our data engine, model training parameters, and further analysis of our experimental results.

## A.2 MAD FRAMEWORK DESIGN AND DEPLOYMENT

### A.2.1 CLASSIFICATION AND COMPLETENESS OF BIAS TYPES

The six types of bias proposed in our "Shortcut Cookbook" (Figure 5 of the main paper) are designed to cover the key failure modes in shortcut learning comprehensively. The completeness of this classification is demonstrated by its correspondence to established bias categories in existing research:

- **Factual Reasoning**: Addresses biases arising from a model's failure to reason about factual relationships between correctly identified objects. This is a form of *vision reasoning bias*.
- **Counterfactual Verification**: Targets the model's reliance on stereotypes (e.g., bananas are yellow) even when contradicted by visual evidence. This corresponds to tackling *background bias* or attribute-based shortcuts, where the model must verify if the core object identification holds true across counterfactual attributes.
- **Small, Edge, OoD Objects**: Corresponds to *local cue bias*, where the model over-relies on salient objects and ignores smaller, peripheral (edge), or out-of-distribution (OoD) ones that are critical for correct understanding.
- **Distorted, Occluded, Rotated Text**: Addresses failures in Optical Character Recognition (OCR), a specific form of perceptual bias related to text.
- **Size, Carry, Spatial Relationships**: Pertains to biases from *co-occurring objects*, where the model misinterprets the interaction, relative size, or position of objects.
- **Counting**: Addresses a specific numerical stereotype where the model associates certain scenes with fixed number ranges (e.g., a station platform always has 3-5 pillars) due to frequent co-occurrence in training data.

We validated this taxonomy through extensive error analysis on challenging datasets such as VQA-CP, VQA-CE, and GQA-OOD, ensuring our categories are both comprehensive and representative of common MLLM failure modes.

### A.2.2 DEPLOYMENT COST AND PERFORMANCE TRADE-OFF

The MAD framework adopts a "test-time scaling" paradigm, introducing additional specialized critic agents during inference to reflect on and correct biased responses from the base model. The associated computational overhead is a strategic trade-off for higher response quality, not an inherent inefficiency. As shown in Table 3, we can effectively balance cost and accuracy by setting a limit on the maximum number of sequential agent calls, demonstrating the framework's flexibility.

Furthermore, we investigated the impact of the critic agents' calling order. We found that prioritizing reasoning-focused agents (i.e., Factual Reasoning and Counterfactual Verification) reduces computational cost due to shorter reasoning chains but yields only marginal gains in debiasing performance compared to a fully dynamic routing strategy.

Table 3: Cost-Accuracy Trade-off under Different Agent Call Limits

| Max Routing Count | $\leqslant 2$ | 3 | 4 | 8 |
|---|---|---|---|---|
| **Avg. Cost (tokens)** | 327 | 510 | 594 | 906 |
| **Avg. Steps** | 1.96 | 2.48 | 2.86 | 4.83 |

### A.3 THE FULL PROCESS OF DATA ENGINE CONSTRUCTION

To train our bias-disentangling router agent and the specialized dimension critic agents, we designed a powerful, automated data engine. The process is as follows:

#### A.3.1 CONSTRUCTION OF DEBIASING PROCESS DATA (CRITIC AGENT TRAINING DATA)

Based on the biased samples identified by the router, we constructed targeted debiasing Chain-of-Thought (CoT) data to train our six types of Dimension Critic Agents:

- **For Cognitive Biases (Factual Reasoning, Counterfactual Verification)**: These "Cognitive Agents" were trained on data generated by a sophisticated process. We used an annotation model (GPT-4o) to first generate fine-grained textual descriptions for every object in the image. The model was then prompted to perform joint reasoning over the original image, the query, the biased baseline answer, and these newly generated descriptions. This process encouraged the model to verify inter-object relationships and underlying factual semantics, correcting its initial shortcut-based reasoning. The resulting successful debiasing trajectories were added to our training set.

- **For Visual Perception Biases (Small/Edge/OoD Objects, OCR, Counting, etc.)**: To train these "Perception Agents," we integrated a **Visual Model Zoo** composed of specialist vision models, including CLIP for classification, SAM 2 for segmentation, and Depth Anything V2 for 3D estimation. We emphasize that we did not rely solely on GPT-4o for annotation. Instead, this powerful vision ensemble pre-generated accurate visual information (e.g., object locations, categories, segmentation masks), which significantly reduced the likelihood of the subsequent annotator LLM producing biased or hallucinatory content. Following the Visual-CoT methodology, the outputs from these vision experts, along with dimension-specific prompts, were iteratively fed to the annotation MLLM until a verified, corrected answer was produced.

Through this automated engine, we curated approximately 50k high-quality, multi-step debiasing CoT data points from an initial pool of 90k processed biased pairs. This final dataset is named the Multi-Dimensional Debiasing Dataset (MD$^3$).

#### A.3.2 FINAL MODEL TRAINING

The high-quality reasoning chains in the MD$^3$ dataset, which capture the cascaded correction trajectories of our multi-agent system, are used directly for Supervised Fine-Tuning (SFT) of a target MLLM. The goal is to distill the collaborative, multi-dimensional reasoning process of the MAD system into a single, more robust model.

- **Data Mixture**: During instruction tuning, the 50k samples from our MD$^3$ debiasing dataset are mixed with a general-purpose instruction fine-tuning set in a 1:3 ratio to maintain the model's general capabilities while enhancing its robustness.

- **Two-Stage Fine-Tuning**: Following the LLaVA-v1.5 methodology, we conduct a two-stage fine-tuning process. In the first stage, only the visual connector (the projection layer) is trained, using a learning rate of $1 \times 10^{-3}$. This aligns the vision encoder with the frozen LLM. In the second stage, we fine-tune both the LLM and the visual connector for one epoch, with learning rates of $2 \times 10^{-6}$ for the LLM and $1 \times 10^{-5}$ for the connector, respectively. All models were trained on 8 NVIDIA A100 GPUs.

### A.3.3 DETAILS OF CLASSIC VISUAL DEBIASING CLASSIFICATION TASKS

We validated the core principles of MAD on classic vision debiasing benchmarks using a ResNet-50 backbone pre-trained on ImageNet as the visual feature extractor. In this simplified setting, the MAD framework is adapted by implementing agents as simple linear correctors that focus on either class or semantic features. The collaborative process is simulated as follows:

- **Optimizer**: SGD with an MSE loss was used for training the base and each corrective agent.
- **WaterBirds**: Learning rate of $1 \times 10^{-4}$; each agent was trained for 200 epochs. Convergence was reached after 9 agents were sequentially added.
- **UrbanCars**: Learning rate of $5 \times 10^{-4}$; each agent was trained for 20 epochs. Convergence was reached after 62 agents were added.
- **CelebA**: Learning rate of $1 \times 10^{-2}$; each agent was trained for 20 epochs. Convergence was reached after 65 agents were added.

Unlike the sequential, cascaded workflow in the MLLM setting, inference for these classic tasks is parallelized: the biased base model and all corrective agents produce outputs simultaneously, which are then integrated to form the final, debiased prediction.

### A.4 OPEN-SOURCE COMMITMENT

We are committed to open-sourcing all related code and resources to facilitate future research. This includes modules for our $MD^3$ dataset construction, model training scripts, and inference code for the complete MAD framework. Each component will be implemented as an independent, well-documented interface with specified inputs to ensure ease of use and reproducibility.

### A.5 RELATED WORK ON DEBIASING

Various approaches exist for learning debiased models using different levels of bias information. Methods guided by explicit bias supervision Sagawa et al. (2020a); Goel et al. (2021); Tartaglione et al. (2021); Cheng et al. (2021) include adding a bias prediction branch and utilizing techniques like mutual information minimization and ensemble learning to reduce bias Ganin et al. (2016). These techniques are demonstrated in works by Kim *et al.* Kim et al. (2019), Li and Vasconcelos Li & Vasconcelos (2019), Clark *et al.* Clark et al. (2019), and others Wang et al. (2020). When explicit supervision is limited, leveraging bias prior knowledge allows constructing modules that address specific bias types Hendricks et al. (2018); Geirhos et al. (2019); Li et al. (2021); Cadène et al. (2019); Arjovsky et al. (2019), as shown by Wang *et al.* Wang et al. (2019) and Bahng *et al.* Bahng et al. (2020) Finally, debiasing through intrinsic bias properties exploits inherent bias characteristics without needing explicit guidance or prior knowledge, using strategies such as two-branch training and bias-contradictory augmentation Darlow et al. (2020); Huang et al. (2020); Zhu et al. (2021); Liu et al. (2021); Kim et al. (2022); Kirichenko et al. (2023b), highlighted by Nam *et al.* Nam et al. (2020) and Lee *et al.* Lee et al. (2021) These collective efforts illustrate diverse ways to tackle bias in machine learning models.

