# OpenReview forum: "Dimensional Debiasing via Multi-Agent Correction"
_ICLR.cc/2026/Conference — ICLR 2026 Conference Withdrawn Submission_

### Official Review · Reviewer_fDRT · 2025-10-31

**Soundness:** 2
**Presentation:** 2
**Contribution:** 2
**Rating:** 2
**Confidence:** 4

**Summary:**

This paper proposes MAD (Multi-Agent Debiasing), a framework to reduce shortcut biases in multimodal large language models (MLLMs). The authors argue that MLLMs often rely on spurious correlations (e.g., clocks reading 10:10 regardless of hand positions) due to biased training data and limited cross-dimensional reasoning. MAD introduces a set of specialized “dimension critic” agents that inspect a model's initial answer, identify the type of shortcut error, and iteratively refine it. A router agent dispatches queries to six critic types covering biased cognition (factual reasoning and counterfactual checks) and limited perception (object recognition, OCR, spatial reasoning, and counting). This process is used both at inference time and as a data engine to build a new dataset, MD3, consisting of ~50k debiased reasoning chains.

Models fine-tuned on MD3 show consistent gains on spurious-correlation and robustness benchmarks, including VQA-CP, VQA-CE, GQA-OOD, OCRBench, and RealWorldQA, with reported improvements up to ~4–6% absolute and larger gains on the MD3 suite.

**Strengths:**

- Intuitive "cookbook" for agent specialisation, which splits errors into 1) Biased Cognition (Errors from the LLM component) and 2) Limited Perception (Errors from the Vision component).

**Weaknesses:**

- The technical novelty appears limited, with the primary contribution centered on a specific multi-agent prompting system and a generated dataset. The paper would benefit from more rigorous justification and analysis of the engineering choices, as well as deeper scrutiny of the resulting dataset.
- Key details about dataset construction are missing. Section 3.3 does not clearly describe how the system verifies that (1) the true shortcut bias has been correctly identified, and (2) the debiasing chain resolves it. Given the scale, this process is likely automated, yet there is no explanation of the validation or iteration procedure. Quantitative indicators of data quality (for example, agreement with other models or human-in-the-loop checks) would strengthen confidence in the dataset.
- The reported gains are uneven. In Table 1 under “General Knowledge, Comprehension, and Reasoning,” 8 of 9 model-dataset combinations show less than a 1% improvement and 4 result in declines. The headline ~4% improvement “on challenging benchmarks” appears driven largely by MD³-related settings, and the improvements are far smaller elsewhere. A more balanced discussion of performance variation across settings, along with clarification of which benchmarks constitute the reported gains, is needed.
- The introduction of sufficient, weak-sufficient, and unrelated “dimensions” offers useful intuition, but the formal definition in Definition 1 is imprecise and mixes set-based and probabilistic concepts. The authors should consider expressing these definitions probabilistically to improve conceptual clarity.

**Questions:**

1) The sentence "Both successfully corrected and persistently biased examples are used as positive and negative samples to further train the Router Agent, improving its diagnostic capabilities over time." is confusing. From my understanding, the router agent is GPT-4o-mini - and this is not being trained. Could you please clarify how the diagnostic capabilities improve over time?

2) The sentence "Our final MD3 dataset consists of approximately 50k debiased reasoning chains, generated from an initial pool of 90k instruction prompts" requires clarification. What is the source of the 90k instruction prompts to begin with? And, as in the weaknesses, what is the specific filtering procedure for obtain 50k chains from these prompts? This is particularly important to clarify to deem the degree of overlap between the MD3 training set and the downstream tasks used to evaluate the debiased MLLM.

---

### Official Review · Reviewer_TGLd · 2025-10-31

**Soundness:** 2
**Presentation:** 3
**Contribution:** 2
**Rating:** 2
**Confidence:** 4

**Summary:**

This work proposes an agent-based system for reducing common errors in MLLM-based VQA systems.
A taxonomy of six common types of pitfalls or errors is proposed, along with an agentic framework for addressing each error type during inference. A dataset of reasoning traces from running this system on multiple datasets is provided.
Mainly, this work is evaluated by first collecting this dataset, and then fine-tuning popular MLLMs like Llava and Llama. There are also ablation studies regarding cross-validation on the agent pool.

**Strengths:**

* This work does a good job at executing on a reasonable research pipeline whereby a more refined dataset is collected and improved results are obtained by training on this data

* The paper is fairly well written and easy to follow

* There are some substantial gains made on certain datasets, especially less popular datasets that test specific biases.

**Weaknesses:**

I have issues with both the significance and the novelty of this approach.


Significance:

* Ostensibly, the goal here is to reduce bias and errors via this multiagent-style reflection and correction setup. The framework is informed by a 6-part taxonomy of errors, but I have a hard time believing that this is in any way a comprehensive categorization of common pitfalls. L796 says "We validated this taxonomy through extensive error analysis on challenging datasets such as [...]", details on this process are critical to understanding this approach yet I don't see them.
* I'm not convinced that we need this multiagent setup, what are the relative strengths of the models in the pool used to generate the data (see Q2). It seems like we could just pick one model like Qwen and use different prompts. Since there is no evaluation of the relative gain by using different models or with / without tools, many questions are left open.
* There is no evaluation of the router itself, which is a crucial part of the design.
* I think there's some overindexing on the "10:10 dilemma" here, its rather basic ML that models will learn spurious features in their training data. But these features can (and will be) anything, so reducing down to 6 dimensions seems fraught.


Novelty

* In the end, this is a multiagent system that is used to label data that we can use to train MLLMs. One high-profile work here is UltraFeedback [1], which mainly only differs in that there are images here. I don't believe the agentic system itself or the proposed error taxonomy are enough to clear the bar here.

**Questions:**

1) How was the taxonomy of errors validated? How many of each error occur in each dataset examined? Please show that the proposed error categories are comprehensive.

2) What is the model pool used in this work? I see Qwen-vl, Internvl, and Phi-3.5 used in the figures but are they chosen for any specific reason?

3) What are the different agent prompts?

4) Why tools and different agents? If we're already using GPT4o / closed foundation models, then why not just let it do everything? I can think of reasons but they need to be stated with evidence. L839 states "We emphasize that we did not rely solely on GPT-4o for annotation". I am not convinced this is a good choice for the stated goals.

---

### Official Review · Reviewer_8heR · 2025-11-01

**Soundness:** 2
**Presentation:** 1
**Contribution:** 2
**Rating:** 2
**Confidence:** 4

**Summary:**

This paper aims to address the problem of MLLMs learning shortcuts (i.e., dataset biases) due to a lack of adequately representative samples in their training datasets. A framework called Multi-Agent Debiasing (MAD) is proposed to correct such shortcuts in reasoning through specialized agents which verify model responses across multiple different dimensions of common failure modes. Specifically, a router is used to route responses through various agents depending upon the types of possible shortcut errors. Each agent iteratively refines the response that was output by the prior agent in a cascading manner. The agents are designed based on a taxonomy of common failure modes in MLLMs which is introduced. The MAD approach is used to construct a dataset called MD3, which is then used to finetune MLLMs to mitigate their reliance on shortcuts. Experiments are conducted using 3 MLLMs and a range of different benchmark datasets.

**Strengths:**

1. The problem of mitigating shortcuts / spurious correlations learned by MLLMs is important and has broad applicability across multimodal reasoning tasks.
2. Plenty of helpful illustrative examples are provided throughout the paper
3. Ablation studies are conducted which indicate that increasing the number of agents improves performance

**Weaknesses:**

1. The paper has presentation and clarity issues. For example, L137: what is the motivation for claiming a uniform distribution here? L142-144: I do not think "sufficient" is an appropriate term to use here w.r.t. the example. An object being boat-shaped might be a necessary condition for a lifeboat, but is not sufficient (i.e., there are many boat-shaped objects which are not lifeboats). L216: the title above figure 4 is nonsensical. L325: the header of the table is misaligned with the columns.
2. One downside of the proposed approach is that the router identifies the sequence of agents only after the initial response is generated. If one of the agents introduces its own biases/shortcuts, there is no mechanism to identify this and reroute the output to an appropriate agent for correction.
3. It's unclear how the proposed bias taxonomy ("shortcut cookbook") was developed and validated. Examples are provided in Figure 5, but it's hard to tell how representative these failure cases actually are in practice. L314 states that "these failure patterns are consistent across various state-of-the-art MLLMs and datasets" - was this quantified in any way via automated and/or human analysis?
4. Models trained with MAD seem to offer little or no improvement on general knowledge, comprehension, and reasoning - is this expected? It seems like there could still be cases of biases/shortcuts learned by models which could impact these tasks.
5. MAD training seems to decrease performance in some cases (e.g., Hallusion for LLaVA-Llama-3-8B) and does not outperform other debiasing methods in some settings (Figure 6a). Better discussion of how these results still support the proposed method would be helpful.
6. The main results (Table 1) are lacking a natural baseline such as training on another existing dataset rather than your proposed MD3 dataset. I wonder if we would still see improvements on these benchmarks simply by training on more data that isn't produced by MAD.

**Questions:**

See questions in my weaknesses articulated above.

---

### Official Review · Reviewer_Xb42 · 2025-11-01

**Soundness:** 3
**Presentation:** 3
**Contribution:** 2
**Rating:** 2
**Confidence:** 3

**Summary:**

MAD (Multi-Agent Debiasing) is a framework that reduces shortcut biases in Multimodal LLMs by leveraging multiple specialized “critic” agents to detect and correct biased responses. A Router Agent identifies the bias type and delegates the correction to appropriate Dimensions. These agents refine the response through cascaded reasoning, often supported by vision tools, to produce a debiased explanation. The resulting corrected reasoning chains form the MD3 dataset, which is then used to fine-tune models so they learn debiased reasoning internally. MAD improves MLLM robustness and accuracy across several bias-sensitive benchmarks.

**Strengths:**

1.  MAD corrects reasoning chains, teaching the model how to avoid shortcuts rather than merely suppressing them.
2. Multi-Agent Specialization Improves Correction Quality
3. Scalable Automatic Data Generation (MD3 Dataset)
4. Fine-tuning with MD3 leads to consistent improvement across general and bias-sensitive multimodal benchmarks, showing that MAD enhances both robustness and overall reasoning capability.

**Weaknesses:**

1. Potential Bias Propagation in MD3 Data Generation:
While the automatic construction of the MD3 dataset via multi-agent correction is a notable strength, it also introduces a risk of bias amplification from the supervising agents themselves. Because the Router and Critic agents are LLM-based, their own cognitive or perceptual biases may inadvertently shape the “debiased” reasoning chains, potentially reinforcing rather than mitigating certain shortcuts. Or could do introducing another type of bias. The paper would benefit from a clearer discussion on quality control mechanisms, such as cross-model validation, human-in-the-loop sampling, or calibrated uncertainty measures, to ensure that the dataset does not inherit systematic bias from the agents.

2. Limited Novelty Beyond Data Generation and Fine-Tuning:
Although the paper presents a well-engineered debiasing pipeline and demonstrates strong empirical performance, the core contributions center primarily around dataset construction and subsequent fine-tuning. The conceptual novelty may appear somewhat incremental, as the method largely leverages established components—multi-agent prompting, tool-assisted reasoning, and standard instruction-tuning—to achieve debiasing. The work would be strengthened by deeper theoretical insights or more principled modeling innovations that go beyond data augmentation, such as formal guarantees on bias reduction, theoretical analysis of multi-agent correction dynamics, or mechanisms enabling intrinsic bias resistance without reliance on curated data.

**Questions:**

Please refer to the weakness.

---

### Note · Authors · 2026-01-24

I have read and agree with the venue's withdrawal policy on behalf of myself and my co-authors.